# Multimodal Neuromonitoring and Neurocritical Care in Swine to Enhance Translational Relevance in Brain Trauma Research

**DOI:** 10.3390/biomedicines11051336

**Published:** 2023-04-30

**Authors:** John C. O’Donnell, Kevin D. Browne, Svetlana Kvint, Leah Makaron, Michael R. Grovola, Saarang Karandikar, Todd J. Kilbaugh, D. Kacy Cullen, Dmitriy Petrov

**Affiliations:** 1Center for Neurotrauma, Neurodegeneration & Restoration, Corporal Michael J. Crescenz Veterans Affairs Medical Center, Philadelphia, PA 19104, USA; 2Center for Brain Injury & Repair, Department of Neurosurgery, Perelman School of Medicine, University of Pennsylvania, Philadelphia, PA 19104, USA; 3University Laboratory Animal Resources, Department of Pathobiology, School of Veterinary Medicine, University of Pennsylvania, Philadelphia, PA 19104, USA; 4Department of Bioengineering, School of Engineering and Applied Science, University of Pennsylvania, Philadelphia, PA 19104, USA; 5Department of Anesthesiology and Critical Care Medicine, Perelman School of Medicine, University of Pennsylvania, The Children’s Hospital of Philadelphia, Philadelphia, PA 19104, USA

**Keywords:** swine, acquired brain injury, traumatic brain injury, coma, disorders of consciousness, subarachnoid hemorrhage, neurocritical care, neurointensive care unit, multimodal neuromonitoring, translational neurotrauma

## Abstract

Neurocritical care significantly impacts outcomes after moderate-to-severe acquired brain injury, but it is rarely applied in preclinical studies. We created a comprehensive neurointensive care unit (neuroICU) for use in swine to account for the influence of neurocritical care, collect clinically relevant monitoring data, and create a paradigm that is capable of validating therapeutics/diagnostics in the unique neurocritical care space. Our multidisciplinary team of neuroscientists, neurointensivists, and veterinarians adapted/optimized the clinical neuroICU (e.g., multimodal neuromonitoring) and critical care pathways (e.g., managing cerebral perfusion pressure with sedation, ventilation, and hypertonic saline) for use in swine. Moreover, this neurocritical care paradigm enabled the first demonstration of an extended preclinical study period for moderate-to-severe traumatic brain injury with coma beyond 8 h. There are many similarities with humans that make swine an ideal model species for brain injury studies, including a large brain mass, gyrencephalic cortex, high white matter volume, and topography of basal cisterns, amongst other critical factors. Here we describe the neurocritical care techniques we developed and the medical management of swine following subarachnoid hemorrhage and traumatic brain injury with coma. Incorporating neurocritical care in swine studies will reduce the translational gap for therapeutics and diagnostics specifically tailored for moderate-to-severe acquired brain injury.

## 1. Introduction

Acquired brain injury—event-related brain damage, such as traumatic brain injury (TBI) or stroke—is frequently debilitating when not outright fatal, and outcomes are often dependent on neurocritical care. TBI is a leading cause of death and disability, with global incidence in approximately 69 million people per year [1]. The Institute for Health Metrics and Evaluation’s Global Burden of Diseases, Injuries, and Risk Factors study found that, in 2016, there were approximately 27 million new TBIs that required hospital care, which likely skews toward the “moderate-to-severe” due to the oversimplified nature of the injury severity spectrum [2]. That study also found that there were approximately 12 million new stroke cases in 2019, half of which were fatal, making it the second leading cause of death worldwide [3]. Ischemic stroke accounts for the majority of cases, while subarachnoid hemorrhage (SAH) accounted for approximately 10% [3]. Acute brain injury is also associated with increased risk for developing dementia and neurodegenerative diseases such as Alzheimer’s and chronic traumatic encephalopathy [4,5,6,7,8].

Due to low prognostic accuracy and a paucity of treatment options for acquired brain injury, there can be a tendency to give in to nihilism when making care decisions in the neurointensive care unit (neuroICU). Indeed, the leading cause of death in the neuroICU is withdrawal of care [9,10,11]. However, recent clinical studies have shown that the potential for recovery is greater than expected, revealing that the prevalent negative prognostic bias is unwarranted [11,12,13,14,15]. In addition, an exhaustive meta-analysis recently found that neurocritical care significantly improves outcome for adults following brain injury [16]. As the neurotrauma field dismisses nihilism and moves forward with renewed determination, the preclinical study of neurocritical care would be an invaluable paradigm to improve clinical prognostic accuracy and offer a viable path for discovery and translation of effective treatments. Indeed, the Neurocritical Care Society’s Curing Coma Campaign has repeatedly called for bidirectional translation in preclinical modeling [10,17,18]. Furthermore, the mounting evidence of the essential role neurocritical care plays in improving neurological outcomes after moderate-to-severe TBI dictates that any preclinical translational work should strive to incorporate neurocritical care techniques and paradigms. 

Swine are ideal subjects for the preclinical study of acquired brain injury and neurocritical care. Since swine are large mammals, human neuromonitoring equipment is directly compatible with them, greatly increasing clinical relevance. Compared to ubiquitous small animal models, swine have large gyrencephalic brains with high white:gray matter ratios, similar to what is found in humans (60:40 in swine and humans versus 14:86 in rats and 10:90 in mice), and these physical properties have major implications for injury mechanisms and pathological manifestations [19,20,21,22]. Indeed, swine enable the study of white matter damage and effects on connectivity due to stroke and TBI. Furthermore, the meningeal subarachnoid space around the swine brain is similar to that of humans and, as such, allows for blood and clot accumulation similar to what is observed clinically with SAH [23,24,25]. These and other advantages of the use of swine in stroke research have been explored extensively in a comprehensive review from Melià-Sorolla and colleagues [26].

In addition to the factors that make swine an ideal translational model for stroke, modeling human closed-head TBI presents challenges that are uniquely addressed by swine. In humans, TBI begins with an intense instant in which mechanical forces are exerted on the brain, followed by days, weeks, or even years of secondary injury mechanisms. Different mechanical forces can result in dramatically different injuries and injury manifestations. The loading mechanisms that generate the mechanical forces of human TBI include impact-loading, which can result in focal lesion (usually cortical) with a gradient of pathology emanating from it; and rotational acceleration/deceleration-induced inertial loading, which results in diffuse injury to neurons, glia, and vasculature throughout the brain [22,27,28]. These loading mechanisms can occur in combination (e.g., impact causing acceleration or vice versa), and inertial loading due to acceleration often occurs without any significant impact loading, but it is exceedingly rare for impact loading to occur in humans without any resultant head acceleration and inertial loading. Inertial loading is unique to humans and other large animals because the injurious forces caused by rotational acceleration are dependent on the mass of the brain, and therefore even extremely high accelerations generate very little force within small brains [28,29,30]. Given their relatively large brain mass, we can scale up acceleration in pigs to achieve the same forces experienced by humans and even tease apart kinematic elements of the injury (e.g., max acceleration, max deceleration, and jerk) to test their influence on recovery and pathology [31]. The aforementioned white:gray matter ratio and gyrencephalic cortex also influence the distribution and effects of injurious forces generated by inertial loading [22,27]. 

Among the manifestations of human TBI that are specifically due to rotational acceleration, the most obvious is traumatic loss of consciousness (coma) [32,33,34], which is the primary diagnostic for guiding neurocritical triage following TBI in humans. Indeed, in the absence of other major polytrauma, coma duration and severity are the primary criteria for determining whether a TBI patient will enter intensive care, as a score of ≤8 on the Glasgow Coma Scale (GCS) typically mandates intubation. Following the brief mechanical injury, secondary injury mechanisms include ischemia due to increased intracranial pressure (ICP) impairing brain perfusion, as well as a variety of interwoven cell/molecular injury cascades, such as inflammation, excitotoxicity, oxidative stress, and others. Because the ubiquitous impact-loaded rodent models share similar cell/molecular mechanisms of secondary injury with human TBI, these cell/molecular mechanisms have historically been the primary focus of preclinical TBI research. Unfortunately, rodent models cannot recreate the mechanisms (inertial loading) or manifestations of human TBI that are due to rotational loading (e.g., loss of consciousness) [28,29,30]. In addition to an inability to produce traumatic loss of consciousness, without mass lesion or hypoxia, rodent TBI models also do not reproduce the secondary increase in ICP that guides most treatment decisions in the neuroICU [35]. Thus, due to a variety of reasons, in addition to an overreliance on impact-only small animal models, our field has yet to translate any of the therapeutics found to be effective for treating impact-only brain injuries in rodents. Rodent models will always be the foundation of preclinical brain injury research, but for reliable, fail-early, pre-IND/IDE (investigational new drug/investigational device exemption) studies, we must employ large animal models that better replicate the mechanisms and manifestations of the human injury.

The swine model of rotational-acceleration-induced TBI is currently the only preclinical model that scales rotational-acceleration-induced inertial loading to recreate the forces of mild-to-severe human TBI [22,27]. Importantly for our purposes, this model can reliably produce a prolonged coma that would lead to admission to the neuroICU if presented in a human patient [36,37]. Swine models of SAH also provide high-fidelity modeling of the human injury due to the similarities of the subarachnoid space and their high white matter content [23,24,25,26]. However, if we are to recreate the mechanisms and manifestations of the human injury, we must go beyond the inciting incident. The course of both injuries typically takes patients through a neuroICU, where they receive neurocritical care and monitoring, introducing influential variables that can affect the injury course and provide extensive neuromonitoring data for which we are striving to improve prognostic value.

Generally, the only existing treatments for acquired brain injury in the acute/subacute phase involve managing endophenotypes in the neuroICU guided by multimodal neuromonitoring (MMNM). There are medical options for clot clearance within a tight window following ischemic stroke (tissue plasminogen activator or mechanical thrombectomy), as well as limited surgical options for clot clearance following hemorrhagic stroke. Decompressive craniectomy offers a surgical approach to reduce damage from intracranial pressure (ICP) after brain injury, though results of clinical trials that defy straightforward explanation leave some questions surrounding its efficacy unresolved [38,39,40,41]. Beyond that, monitoring and responding to secondary injury processes—including altered ICP and partial brain tissue oxygen (PbtO_2_)—following moderate-to-severe acquired brain injury are central to achieving a positive outcome [16]. It is therefore important to recreate the intensive care environment in a preclinical model to (a) account for and study the primary variables experienced by humans after injury (including the interventions encountered in the ICU); (b) provide clinically relevant neuromonitoring data in preclinical studies; (c) enable translational development of new treatments specific to the neurocritical care space; and (d) provide the human-level care necessary to extend the study period for preclinical moderate-to-severe TBI with coma, which has historically been limited to 8 h due to the need for sophisticated neurocritical care to survive animals beyond this point [36,37]. 

Others have made valuable progress recreating elements of the clinical neurocritical care environment in swine models of brain injury, as displayed in Table 1. These researchers have employed a variety of injury mechanisms, study durations, monitoring modalities, and other data collection techniques [42,43,44,45,46,47,48,49]. Notably, Friess and colleagues utilized a pediatric swine model of rotational acceleration TBI and measured ICP, PbtO_2_, cerebral blood flow (CBF), and microdialysis for lactate:pyruvate ratio (LPR) over a period of 6 h [42,43,44]. Those studies established a correlation between neurocritical care monitoring and pathology and investigated the interplay of vasopressors with CBF and cerebral perfusion pressure (CPP). The pediatric pig model of rotational acceleration TBI is essential to understanding the unique pathophysiology of pediatric TBI and developing treatments that can prevent long-term consequences [50]. In a swine model of SAH, Nyberg and colleagues collected ICP and microdialysis for glucose and LPR. They validated their model by confirming that cerebral ischemia and metabolic changes after SAH were consistent with the clinical condition [45]. To our knowledge, the longest study duration to date was in a swine model of acute subdural hematoma, in which Datzman and colleagues collected ICP, PbtO_2_, and microdialysis for LPR over a period of 52 h [46]. More recent studies include a model of blast TBI in swine that demonstrated increased coagulopathy and ICP after injury [51], as well as a fluid percussion injury model in swine that monitored ICP, along with transesophageal echo, to study left ventricular function [52], and a swine model directly controlling increases in ICP to test the effects on CBF [53]. We sought to integrate the strengths of these innovative studies while drawing on state-of-the-art clinical practice to maximize the translational relevance of this platform. 

Here, we present the development of a comprehensive multimodal neuromonitoring and neurocritical care suite for use with swine—an accepted large animal species commonly used for IND/IDE-enabling studies. We describe in detail the protocols we developed and present individual examples of monitoring and medical management in subjects administered a sham injury, subarachnoid hemorrhage, or moderate-to-severe TBI with coma under these protocols. Through an iterative development approach to systematically optimize medical management and improve survival, developing this paradigm allowed for data collection from swine experiencing TBI with coma that exceeded 8 h for the first time, yielding notable technical advancements and observations.

## 2. Materials and Methods

All procedures were approved by the University of Pennsylvania’s Institutional Animal Care and Use Committee (IACUC) and the Corporal Michael J. Crescenz VA Medical Center in Philadelphia (both AAALAC accredited) and were completed in accordance with the Guide for the Care and Use of Laboratory Animals [54]. All studies were performed in the Porcine Neurointensive Care and Assessment Facility (NCAF) at the University of Pennsylvania, a state-of-the-art swine injury and behavioral assessment facility created in January 2015 to study acute and long-term responses in porcine models of brain injury.

An alphabetical list of abbreviations can be found in the Abbreviations section after the Discussion.

### 2.1. Iterative Technique Development

This study was designed to adapt and optimize clinical multimodal neuromonitoring and neurocritical care techniques for use in swine. As such, iterative changes were made between each subject as part of the development process. Examples of these changes include sampling frequency, processing procedure, and types of biological samples collected; adjustments to sedation and/or ventilation strategies; streamlining surgical procedures to optimize timing; and altering the animal protocol to add drugs for medical management. Therefore, although our monitoring and care was optimized with 3 sham injury animals, 3 experimental SAH animals, and 2 severe TBI animals with coma administered via rapid angular rotational acceleration of the head, the iterative nature of this study makes it difficult to group subjects together in a way that allows for clear summarizations/comparisons or valid statistical testing beyond the intended purposes of creating a comprehensive neurointensive care unit for studying severe brain injury in swine. Instead, we describe in this Methods section the techniques that we developed, and in the Results section, we present general observations for each condition and medical management in individual subjects.

### 2.2. Workflow Summary 

Our workflow included pre-injury surgeries and monitoring, TBI or SAH injuries, and post-injury monitoring and neurocritical care (Figure 1). Female Yorkshire swine (25–30 kg) were induced via ketamine/midazolam (Hospira, Lake Forest, IL, USA), intubated, and maintained under isoflurane anesthesia. Femoral artery and internal jugular vein catheterizations were performed to allow for continuous blood pressure (BP) monitoring and serial blood draws from an arterial line (A-line), as well as drug administration via triple lumen central line. In some cases, a lumbar drain was placed to facilitate cerebrospinal fluid (CSF) sampling. 

For all subjects in the swine neuroICU, a quad-lumen bolt (Hemedex, MA, USA) was secured 1 cm rostral to bregma for placement of a parenchymal ICP probe (Natus, WI, USA), PbtO_2_/temperature sensors (Integra Licox, Princeton, NJ), Spencer depth electroencephalography (EEG) electrode (SD08R-AP58X-000; Ad-Tech, WI, USA), and microdialysis catheter (mDialysis, MA, USA) per institutional clinical paradigm at the University of Pennsylvania. In some cases, a Bowman CBF monitor (Hemedex, MA, USA) was placed in the quad bolt, and a separate burr hole was used for the depth EEG electrode. Likewise, a surface EEG array was placed on the scalp. In the TBI group, the invasive cranial monitors were placed after injury induction to prevent shearing during rapid acceleration/deceleration of the head. In the SAH animals, cranial monitors were placed prior to injury. Animals in the neuroICU were switched from isoflurane to total intravenous anesthesia, utilizing titrations of propofol and fentanyl, delivered via the central line.

TBI was administered in two subjects via the HYGE pneumatic actuator to achieve inertial loading via rapid rotational acceleration of the head in the sagittal plane (113–114 rad/s). Following head rotational TBI, these subjects were moved to the swine neuroICU. SAH was administered in three subjects via injection of autologous blood into the subarachnoid space at the skull base, using a contralateral external ventricular drain catheter placed down to the skull base. Three sham subjects underwent all ICU-related procedures but received neither TBI nor SAH.

Animals were monitored continuously up to 36 h. Neurological assessments were performed during sedation holidays (propofol off, fentanyl low). Electrocardiogram (EKG), blood oxygen saturation (SpO_2_), capnography, BP, ICP, PbtO_2_/temperature, and EEG (depth + scalp) were time-synchronized and continuously recorded with waveform resolution on a Moberg CNS-200 (Moberg Research Inc., Ambler, PA, USA). Arterial blood, CSF, microdialysate, and urine were collected, processed, and frozen for analysis and biobanking. Plasma biomarker assays were run using the Neurology 4-plex B assay on the ultrasensitive Simoa-HDX bead-based immunoassay platform (Quanterix, Billerca, MA, USA). At the end of the experiment, subjects were deeply anesthetized and exsanguinated via cardiac perfusion with normal saline, followed by 10% neutral buffered formalin for tissue fixation. Formalin-fixed, paraffin-embedded brain sections were stained with hematoxylin (Modified Mayer’s Hematoxylin; 22-110-639; Fisher Scientific, Hampton, NH, USA) and eosin (Eosin Y—aqueous; 6766009; Fisher Scientific, Hampton, NH, USA) (H&E) and immunostained for amyloid precursor protein (APP; mouse; MAB348; 1:80,000; Millipore Sigma, St. Louis, MO, USA) and ionized calcium binding adaptor molecule 1 (IBA1; rabbit; 019-19741, 1:4000; Wako, Richmond, VA, USA) with DAB (3,3′-Diaminobenzidine; SK-4100; Vector Labs, Newark, CA, USA) secondary staining for colorimetric microscopy. Staining for APP reveals white matter pathology, as APP accumulates in damaged or degenerating axons. Staining for IBA1 reveals the morphology, location, and number of microglia in a given brain area, which provides information on the nature and severity of the neuroinflammatory response to injury. Entire sections were scanned using an Aperio CS2 digital slide scanner. Detailed staining and pathological scoring methods can be found in Grovola et al. (2021) [55].

### 2.3. Induction and Line Placement

Prior to injury, animals were anesthetized, and indwelling catheters were placed for repeated blood draws and mean arterial pressure monitoring. Pigs were induced with an intramuscular injection of midazolam (0.4–0.6 mg/kg) and ketamine (20–30 mg/kg), intubated, and maintained on isoflurane (1–5%). Glycopyrrolate (0.01–0.02 mg/kg) or atropine (0.02–0.05 mg/kg) was used to mitigate excessive secretions during intubation. Animals were continuously monitored, and anesthetic levels were adjusted as needed throughout the procedure to maintain a plane of anesthesia, ensuring that the SpO_2_, heart rate, and respiration rate were within acceptable ranges. Thermal support was provided using blankets, a BairHugger warmer, or a HotDogger unit. Temperature was monitored continuously throughout the procedure. For all surgeries, hair at the site was clipped prior to aseptic skin prep. The surgeon wore a cap, booties, a mask, sterile gloves, and a sterile gown. The surgery was performed aseptically, following the IACUC Guidelines for USDA species’ survival surgery.

For the jugular/cephalic vein catheterization (central line), the pig was placed in dorsal recumbency, and Bupivacaine (1–2 mg/kg) was injected subcutaneously over the identified incision site prior to incision. An incision was made, and then the subcutaneous tissue and cutaneous colli muscle were dissected, identifying the vein. The vein was retracted and then catheterized. The catheter was secured in the vessel with silk ties. The catheter was then passed via a sterilized trocar tunneled through the subcutaneous tissues exiting the skin on the dorsum of the neck; alternatively, the exteriorized catheter could be exited through the incision and taped down. The original incision was closed in multiple layers (muscle, subcutaneous, and buried skin), using absorbable or nonabsorbable sutures. If unsuccessful at achieving access, the procedure was performed on the contralateral side. This access was used for drug delivery, including continuous rate infusion (CRI) anesthesia in the neuroICU.

For the femoral artery catheterization (A-line), the pig was placed in dorsal recumbency, and the rear leg was retracted caudally. The medial aspect of the leg (starting at the stifle and extending inguinally) was draped and prepped using chlorhexidine scrub. Under aseptic conditions, the femoral artery was catheterized with an arterial catheterization kit, using the Seldinger technique. This access was used to record the mean arterial pressure and draw samples. The catheter was flushed with saline at the time of each blood draw. 

For the placement of the lumbar drain, the pig was placed in a lateral recumbent position. The hair over the lower lumbar spinous processes was clipped, and the skin was prepped aseptically with repeated surgical scrubs of chlorhexidine and betadine. Under aseptic conditions, a Touhy needle was inserted in the midline and through the interspinous space at the L5–L6 region. Once the thecal sac was entered—confirmed by CSF flow—the Tuohy needle stylet was removed, and the lumbar drain catheter was threaded through the Tuohy needle. The drain was then secured in place and attached to an external collection system.

### 2.4. Rotational Traumatic Brain Injury (TBI)

Following induction, intubation, and line placement, the subjects—still maintained under isoflurane via a portable anesthesia cart—were covered with a blanket and/or Bair-Hugger unit for thermal control and moved to an adjacent procedure room in the NCAF, where brain trauma could be induced via head rotational acceleration. The anesthesia level, heart rate, respiration rate, blood oxygen saturation, and temperature were continuously monitored. The level of anesthesia was gauged by assessing the animals’ jaw tone, as well as a strong pinch of the ear or the webbing between the toes.

This model produces pure inertial non-impact head rotation in different planes at controlled rotational acceleration levels (thus controlling severity). The powerful HYGE pneumatic actuator device can generate 20,000 kg of thrust in less than 6 ms, resulting in peak angular accelerations of over 300,000 rad/s^2^. Of note, the loading conditions generated by this device closely approximate the conditions of inertial brain injury in humans based on brain mass scaling. While this model is labor-, resource-, and skill-intensive, it is currently the only model that can recreate the inertial loading and head rotational acceleration central to human TBI and is therefore the most clinically relevant animal model of closed-head TBI. The subject’s head was secured to a padded snout clamp, which, in turn, was mounted to the linkage assembly of the HYGE pneumatic actuator device. The padded snout clamp is specially designed to convert the linear motion of the HYGE to an angular (rotational) motion in the sagittal plane. To produce closed-head, diffuse, moderate-to-severe TBI, pure head rotation (approximately 55 degrees in <12 ms) was induced. For each experiment, head angular velocity/acceleration traces were recorded. The peak rotational velocities of the two subjects included in this study were 113 and 114 rad/s. Immediately following rotation, animals were detached from the device, assessed for apnea, examined for any injury to the mouth/snout (none was found), and transferred to the swine neuroICU in an adjacent procedure room in the NCAF. Buprenorphine SR was provided when recovery was sufficient to graduate from the neuroICU.

### 2.5. Multimodal Neuromonitoring (MMNM)

After induction of anesthesia and placement of lines—and in the case of the TBI group, following induction of injury—animals were transferred to the swine neuroICU for placement of cranial probes, monitoring, and critical care. A summary illustration of the lines and probes we utilize in the swine neuroICU can be found in Figure 2. 

Subjects were connected to a clinical ICU vital sign monitor (Phillips, Netherlands) for continuous monitoring of the heart rate, arterial and non-invasive BP, oxygen saturation, and EKG tracing. The temperature was continuously monitored via a rectal thermometer probe. End-tidal CO_2_ (EtCO_2_) was monitored at the junction of the breathing tube and endotracheal tube. Cranial monitor placement was then performed aseptically, following the IACUC Guidelines for USDA species’ survival surgery. After a left paramedian incision, a small burr hole was made above the frontal cortex, 5–15 mm paramedian and 5–10 mm cranial of the coronal suture. A quad lumen bolt to house several neuromonitoring modalities was placed into the burr hole. Specifically, an ICP probe (Natus, WI, USA) was placed in the brain parenchyma approximately 1–1.5 cm deep to reach the junction of cortex and subcortical white matter. A PbtO_2_ and temperature probe (Integra Licox, Princeton, NJ, USA) was placed in the subcortical white matter, in combination with a Bowman CBF monitor (Hemedex, MA, USA). A microdialysis bolt catheter with either a standard 20 kD pore size (8002823E; mDialysis, MA, USA) or a high-pass 100 kD pore size (8050194A; mDialysis, MA, USA) was placed 10 mm deep to obtain extracellular fluid solutes for metabolic analysis. The microdialysis catheter was perfused with lactated ringers at low flow (1 µL/min), with dextran added in the case of the high pass membrane. A 1 mm burr hole was made posterior to the bolt for implantation of an 8-lead Spencer depth electrode for continuous EEG monitoring across 17 mm of cortical layers (SD08R-AP58X-000; Ad-Tech, WI, USA). The incision was sutured closed around the bolt housing. Once the bolt and sensors were placed and the animals were stable, they were transitioned to CRI intravenous propofol (5–20 mg/kg/h) anesthesia and fentanyl (0.1–100 µg/kg/h) analgesia for the duration of their time in the neuroICU. A basic EEG surface array was placed over the frontal lobes comprising two electrodes over each hemisphere with grounding electrode. EEG electrodes were routed through a standard Moberg headbox EEG array. All monitors were connected to a Moberg ICU monitor for time-synced recording of monitoring data, vital sign data (Phillips ICU monitor), and EEG (Figure 3).

### 2.6. Administration of Subarachnoid Hemorrhage (SAH)

SAH injury was administered following admission to the neuroICU and placement of lines and cranial monitors. An incision was made over the left frontal lobe, mirroring the placement of the quad lumen bolt. A contralateral burr hole was made at the same landmarks. An external ventricular drain (EVD) catheter was augmented to have a single opening at the distal end. The drain was inserted perpendicular to the outer table of the skull until it reached the skull base. The inner stylet was withdrawn. CSF was carefully withdrawn, confirming placement in a basal cistern. Arterial blood was collected from the arterial line and injected under pressure through the external drain into the basal cisterns. The injection was stopped when cerebral perfusion pressure, as measured by the difference between mean arterial pressure (MAP) and ICP, was 0 mmHg (requiring injection of approximately 10 mL of autologous blood). After the injection was halted, the catheter was withdrawn, the wound was irrigated, and the skin was closed with running nylon suture. 

### 2.7. Neurocritical Care

Neurocritical care in the pigs was modeled after clinical practice, and therefore it was primarily focused on maintaining brain perfusion. This is achieved by monitoring cerebral perfusion pressure (CPP), which is the difference between MAP and ICP. MAP must be high enough to overcome ICP in order to provide sufficient brain perfusion. For adult swine and humans, we seek to maintain CPP above 60 mmHg. The focus on CPP is primarily due to the prevalence of increased ICP observed after brain injury. For that reason, treatment in humans and in our pig model usually involves maintaining ICP below 20 mmHg through a combination of ventilation management, sedation, and hyperosmotic therapy. While increased ICP after brain injury is common, it is also possible to encounter MAP below the optimal range of 80–100 mmHg (also consistent with target values for humans), in which case application of a vasopressor such as norepinephrine may be needed. The focus on maintaining brain perfusion is primarily to prevent hypoxia/ischemia, but hypoxia may sometimes be present even when ICP and CPP are normal [56,57]. Observational clinical studies have reported that low PbtO_2_ is common after severe TBI and associated with poor outcome [58,59,60,61,62], while PbtO_2_-directed care is associated with improved outcome [63,64,65,66]. The BOOST-II clinical trial was a randomized, controlled study conducted in 119 severe TBI patients from 10 ICUs designed to test the efficacy of including PbtO_2_, along with ICP, in guiding treatment [67]. They found that monitoring and treating PbtO_2_ significantly reduced total hypoxia burden, and a phase III trial is currently underway to assess the impact on neurological outcome [68]. Therefore, we also sought to maintain PbtO_2_ above 20 mmHg in the swine neuroICU.

Surface and depth EEGs provide extensive data for research applications, and they also allow us to detect seizures that develop due to post-traumatic epilepsy (PTE). We monitored for seizures and managed them with midazolam and propofol. Acquired brain injury also frequently results in medical events extending beyond the central nervous system. Apnea is common and must be detected immediately in order to administer manual resuscitation via bag valve mask until the animal can be placed on the ventilator. Cardiac events (e.g., stress myocardia) may also occur after injury. Monitoring vital signs such as heart rate (HR), SpO_2_, and respiratory rate provides information for maintaining anesthetic depth, along with providing early—or immediate—signs that intervention is necessary. We also measure EtCO_2_, which informs our ventilator settings and can indicate potentially dangerous changes in pH. Those pH changes, as well as metabolic and other perturbations, are also detected via blood gas analysis, which we performed via an i-STAT handheld blood analyzer when we drew blood samples from the A-line. Neurological assessments were performed during regular sedation holidays (propofol off, fentanyl low), using parameters from the human GCS [69,70] and the canine-modified GCS [71,72,73] that we adapted for use in swine in a sternal position, with limited head movement, due to the presence of cranial probes. Our prototype of the swine GCS can be found in the Results section in Table 2, with scoring for a TBI subject. While the animals were fully sedated, we also monitored them for signs of responsiveness, which may include palpebral reflex, anal or jaw tone, EEG activity, or breathing against the ventilator detectable on the EtCO_2_ trace. 

The tests and measurements detailed here are key indicators of potentially life-threatening events that occur after brain injury, but simply maintaining CPP > 60 mmHg, ICP < 20 mmHg, and PbtO_2_ > 20 mmHg is not sufficient to provide the best chance of recovery. Unexpected or subtle medical events often emerge. Individuals must be trained with clinical neurointensivists and veterinarians to provide care in the swine neuroICU, and a neurointensivist and veterinarian should always be available in person or via phone or video chat for consultation.

### 2.8. Sample Collection and Analyses 

Animals were monitored continuously up to 36 h, and all monitoring waveforms that informed clinical care decisions (e.g., ICP, MAP, HR, SpO_2_, capnography, BP, PbtO_2_/temperature, EEG, etc.) were also time synched and recorded via the Moberg ICU monitor for subsequent analyses. We continuously collected cerebral microdialysate while sampling arterial blood (via A-line), CSF (via lumbar drain), and urine (via an aseptically placed urinary catheter) every 2–4 h, and these samples were all processed and frozen for analysis and biobanking. At the end of the study period, subjects were perfusion fixed with 10% formalin, after which brains were extracted and submerged in 10% formalin overnight for complete fixation. Fixed brains were then sectioned coronally into blocks for gross pathology (5 mm blocks with an initial cut immediately rostral to the optic chiasm), after which the blocks were run through an automated tissue processor and finally embedded in paraffin wax for long-term archival preservation and ease of sectioning. A microtome was used to cut 8 µm thick slices from the formalin-fixed, paraffin-embedded (FFPE) brain blocks, and then the slices were mounted on slides for histological analyses. We stained with H&E and immunostained for either amyloid precursor protein (APP; mouse; 1:80,000) or ionized calcium binding adaptor molecule 1 (IBA1; rabbit; 1:4000) with DAB secondary staining for colorimetric microscopy. Entire sections were scanned using an automated Aperio CS2 digital slide scanner (Leica Biosystems, Germany) at multiple magnifications to allow for cloud-based examination of multiple different brain regions across the breadth of the large gyrencephalic brain sections. Detailed staining methods and pathology scoring techniques can be found in Grovola et al. (2021) [55]. Cranial dialysate was assayed for lactate, pyruvate, glycerol, and glucose via an Iscus clinical microdialysis analyzer. Plasma samples were analyzed via the Neurology 4-plex B assay on the ultrasensitive Quanterix Simoa-HDX bead-based immunoassay platform, a kit designed for detecting brain injury biomarkers, such as glial fibrillary acidic protein (GFAP) and ubiquitin carboxyl-terminal esterase L1 (UCHL-1), in human plasma. 

## 3. Results

This study was designed for the iterative development of comprehensive neurocritical care and monitoring in pigs with severe brain injury, and therefore the primary research products are the capabilities that were successfully developed, as reported in the Methods section. While animals received similar injuries, data were not pooled across animals because critical care variables were changed between each animal to facilitate model development. In addition to the techniques that were developed in this study, several useful observations were made, and notable medical events were encountered that were successfully mitigated. In this Results section, we present these observations and medical management reports. 


**Medical management of sham subjects**


Maintaining an uninjured control subject under CRI anesthesia in a neuroICU for extended periods of time represents one of the unique advantages of preclinical research in this area, but it also presents unique challenges. We cannot overstate the importance of gaining experimental insights and precision by gathering invasive multimodal neuromonitoring, biological fluid samples, neurological assessments, and brain tissue from control subjects that experience all experimental variables, except for the injury under study. Monitoring modalities in our three sham animals did not typically deviate beyond acceptable parameters, and blinded histopathology analyses found signs of pathology and inflammation only near sites of cranial probe implantation. However, special care must be taken to maintain sedation in animals that have not sustained any awareness-altering injury. While remaining within the same dose ranges for anesthetic agents as injured subjects on the same protocol, administration rates will need to be increased over time to maintain sedation during wake cycles. Furthermore, it is important to maintain only the doses required for sedation, not only for experimental reasons but also because tolerance can build to anesthetic agents even within a 24 h period. Therefore, we found that it is optimal to employ a gradual increase in administration rates as needed, with later reductions during low arousal periods when possible. 

### 3.1. High EtCO_2_

Even when no injury was administered, the close monitoring of anesthetized subjects remained essential. For example, 12 h after entering the ICU, one sham subject experienced a decreased HR and BP, along with moderately increased EtCO_2_ and ICP. The ventilation rate was increased, and anesthesia was reduced to compensate. The HR and BP remained low but consistent throughout the night. EtCO_2_ remained high (55–65 mmHg) until the I:E ratio was adjusted from 1:2 to 1:1 (reduced EtCO_2_ to <40 mmHg). In the morning, the subject was temporarily removed from the ventilator to gauge autonomous breathing. As expected, the subject breathed autonomously, the EtCO_2_ fluctuated, the HR increased to >120 bpm, the BP spiked to >120 mmHg, and the ICP spiked to 30 mmHg but decreased to normal levels over 10 min. This example serves to illustrate that there can be medical challenges to keeping an animal anesthetized for long periods of time—whether injured or not—and care must be taken.


**Medical management of SAH**


The administration of the SAH injuries occurred after the subjects were stabilized in the neuroICU and baseline samples were collected. Autologous blood was injected into the basal cistern via an EVD implanted to the skull base. When the MAP and ICP reached the same value (CPP = 0), the injections were stopped. This technique reliably recreated the presence of a large volume of blood in the subarachnoid space that was clearly evident during brain extraction and gross pathology, resulting in signs of localized ischemia, as evidenced by H&E staining (Figure 4). Our blinded histopathology analyst correctly noted that the localized ventral appearance of blood, pathology, and inflammation that was observed was not consistent with the pathology from closed-head TBI [55,74,75,76]. As with the sham animals, we sometimes needed to manage rising EtCO_2_ with sedation and ventilator adjustments. Unlike with the shams, we needed to resort to hyperosmotic therapy to manage the increased ICP. In addition, one animal experienced seizures during the administration of the SAH injury, and another, as described below, experienced a major cardiac event nearly 12 h after experimental SAH.

Approximately 11 h and 40 min after SAH administration, one subject experienced a precipitous drop in MAP and CPP, along with loss of peripheral and brain tissue oxygenation. This cardiac event was followed by a large increase in ICP to 49 mmHg, but brain perfusion would have been of little help at the time because there was little-to-no oxygen in circulation. We started a hypertonic saline infusion, and the ICP gradually came down to acceptable levels over the next 30 min, with restoration of PbtO_2_ following the same time course (Figure 5). 


**Medical management of TBI with coma**


Administration of moderate-to-severe head rotational acceleration TBI occurred after lines were placed, but prior to entry into the neuroICU and placement of cranial probes (which would have been damaged and caused major local trauma if present during the head rotational acceleration). The care team must be fully prepared for ICU admission prior to injury, and critical care must begin immediately following injury, as evidenced by the examples below. As with the sham and SAH animals, underlying ventilator and sedation management was key to achieving stability. However, apnea, cardiovascular events, post-traumatic epilepsy, ICP increases, and other factors made medical management of the TBI animals more challenging overall. Fortunately, the management of ICP with sedation and hyperosmotic therapy was effective (Figure 6a). 

The only FDA-approved plasma biomarkers for TBI are glial fibrillary acidic protein (GFAP) and ubiquitin C-terminal hydrolase-L1 (UCHL1) [77,78]. While they have good prognostic value for mild TBI outcome, their value for severe TBI is questionable, and their use in place of a computed tomography scan is very controversial since every bleed that goes undetected could be fatal [79,80,81]. In general terms, plasma concentrations of GFAP and UCHL1 increase acutely following TBI in humans, while neurofilament light (NfL) increases gradually over time [82]. We found that human assay kits were compatible with pig plasma for detection of GFAP, UCHL1, and NfL, and that in a pair of severe TBI pigs, the time course of these plasma biomarkers after injury was in line with what is observed clinically (Figure 6b). 

### 3.2. Cardiovascular Distress and Post-Traumatic Epilepsy (PTE)

Following injury, one subject experienced repeated periods of apnea and an instance of sudden cardiac arrest, requiring manual resuscitation via bag valve mask until transfer to the ICU, where breathing was controlled by the ventilator; during this time, MAP was monitored externally and found to be low (42 mmHg). Norepinephrine administered via a central line at 0.01 mg/h for 5 min corrected MAP (105 mmHg) and produced a brief spike in the HR (180 bpm). After placement of the cranial bolt and switching to CRI anesthesia, the animal was relatively stable for approximately 15 h (Figure 7a). However, approximately 18 h after injury, the animal began experiencing seizures. Initially, a 5 mg bolus of midazolam was sufficient to terminate seizures, as seen in the EEG trace (Figure 7b). Seizures returned throughout the following morning, requiring propofol boluses in addition to midazolam. CRI midazolam at 10 mg/h effectively terminated seizures. Infusion continued throughout the remainder of the 36 h experiment, during which seizures were infrequent and terminated by propofol bolus.

### 3.3. Coma and Wakefulness without Awareness

In a different subject, we observed signs of ventilator desynchrony in the neuroICU approximately 90 min after injury. During their first sedation holiday for neurological assessment at 2 h post injury (Table 2), there was no jaw tone, some reflexive head movement, and a spike in ICP that resolved when stimulation was discontinued. The animal was breathing without the ventilator at 3 h post injury. During the second assessment holiday, we observed immediate hind-limb reactivity, and responsiveness was observed sooner than in the previous assessment. In addition, the ICP increase was also less severe after removing propofol sedation. Therefore, the animal appeared to be stable and emerging from coma by 6 h post injury, so we switched to isoflurane anesthesia and removed instrumentation to exit the neuroICU and return to the home cage for recovery.

Over the next 10 h after anesthesia was discontinued, the animal did not fully regain consciousness but was clearly not comatose. The subject was responsive to tactile stimuli and could briefly sit up with assistance. After 4 h, they were responsive to auditory startle and displayed spontaneous eye blinks. The animal swallowed water from a syringe but did not display any other response to the water, indicating that it may have been reflexive behavior. We observed a stable HR (98–111 bpm) and SpO_2_ (97–99%) during the 10 h in the home cage, but the temperature was elevated, briefly reaching 104.8 °F. Cool wet towels were applied to the feet, followed by a chilled normosol drip via a central line. In accordance with our protocol at the time, the experiment ended when the animal had not recovered volitional mobility or awareness of environment 10 h after leaving the ICU. During postmortem brain extraction, extensive subdural bleeding was observed across the entire surface of the brain and brainstem (Figure 8a,b). Even at this relatively acute timepoint, histopathological analyses revealed a high burden of APP (scored 3 on a 0–3 scale by a blinded analyst), which is indicative of significant diffuse axonal injury throughout the brain, and elevated reactive microglia with short, thick processes that are indictive of a neuroinflammatory response. This pathology was particularly severe in the periventricular white matter dorsolateral to the lateral ventricles and in the fornix/septum pellucidum (Figure 8c–e), as compared to the same areas in the brain of a sham animal (Figure 8f–h). In these areas, differences in APP pathology were stark, with severe axonal damage being apparent in the TBI brain and a near absence of any APP signal in the sham brain. Similarly, IBA1 staining in the dorsolateral periventricular white matter revealed overt tissue damage and dense ameboid microglia along the edge of the ventricle in the TBI brain (Figure 8e). Farther from the edge, the regions of interest magnified in Figure 8e show reactive microglia with a short, stubby process in the TBI brain, while the processes in the sham brain (8h) create a fine network throughout the parenchyma passing in and out of plane to give a somewhat speckled appearance. Enlarged lateral ventricles are also evident in the images from the TBI animal when compared to the sham animal.

## 4. Discussion

To develop appropriate diagnostics and treatments in the preclinical space, the field must strive to model the physiologic and clinical factors that collectively dictate patient outcomes. To that effect, the acute period with the most significant change in physiology and the initiation of the injury cascade, and the subacute period presenting with peak brain swelling and deleterious pathophysiological changes, require the highest-fidelity modeling. The best way to accurately predict the translational potential of newly developed tools and treatments is to not only model the mechanisms/manifestations of the human injury but also the unique environment of the neuroICU in which the patient will be treated. We hope that by addressing both needs, the capabilities developed in this study will help to improve the translational pipeline in neurotrauma.

Swine and human brains share many key characteristics that are not found in small mammals which influence both the mechanisms and consequences of stroke and TBI. In addition to their gyrencephalic cortical architecture and high white matter content, pigs are also more similar to humans than smaller mammals in the anatomy of their limbic, subcortical, diencephalic, and brainstem structures [19,20,21,22,83,84]. These features, along with similarities between human and pig basal cisterns in the subarachnoid space and the aforementioned high white matter content, make the pig an excellent model for SAH [23,24,25]. These anatomical similarities also lead to high-fidelity modeling of human TBI, particularly for rotational acceleration/deceleration-mediated inertial loading, which is dependent on brain size [22,27,28]. Indeed, despite rotational acceleration injury attempts made with the rodent “CHIMERA” model, it is not physically possible to scale up the acceleration high enough (8000%) to recreate the forces of human TBI given the small brain mass of rodents [28,29,30]. Fortunately, some CHIMERA users have dutifully reported failure to reach scaled thresholds for inertial loading, with diffuse pathology clearly emanating from the impact site, but a large swath of the field may still be unaware that the CHIMERA model does not provide any acceleration-induced brain injury in rodents [85,86]. By recreating the mechanism of inertial loading in the pig model, we are able to recreate the manifestations of human TBI—such as loss of consciousness and increased ICP—that require closed-head rotational acceleration acting upon a sufficient brain mass, which cannot be recreated in rodents [27,28,29,30,32,33,34,35]. Rodent models will always be highly relevant and foundational to neurotrauma research given their low cost, time for iteration, well-characterized and readily altered genetics, and significant overlap with secondary cell/molecular injury mechanisms, but there are insurmountable obstacles to direct translation from rodents to humans that are best addressed by filling the gap with pig models. 

Utilizing the monitoring and care capabilities of our reverse-translated neuroICU, we observed and addressed many manifestations of the human injuries being modeled. The collaboration between neurotraumatologists, clinicians, and veterinarians in the administration of this study was vital to guiding the medical management decisions that, in turn, informed algorithms for future work, and this multidisciplinary team will be vital as we further refine our protocols. The inclusion of sham controls emphasized that rising EtCO_2_ can emerge in anesthetized animals on a ventilator even when they are not injured, and if this is not appropriately addressed via sedation, ventilator adjustments, or by taking the animal off of the ventilator, it can have systemic effects that negatively impact brain perfusion. The stress cardiomyopathy that we observed in one of our SAH animals (Figure 5) also commonly occurs in human patients after SAH and is thought to be associated with excessively high catecholamine signaling [87,88,89,90,91,92]. During the stress cardiomyopathy that we observed following experimental SAH, there were instances of artifactual interference that made it appear as if the subject’s heart stopped for up to 30 s. However, while these were artifact and not cardiac arrest, in some instances, cardiac arrest may follow SAH stress cardiomyopathy and is thought to be connected to larger volume bleeds that damage the hypothalamus or brainstem vasomotor areas, though the mechanisms connecting SAH to these potentially fatal cardiac events are not well understood [90,92,93,94,95]. This model seems ideal for testing the dose/response of bleed volume to outcomes after SAH. 

Our plasma biomarker results demonstrate compatibility with the human brain biomarker assay kit and show promising trends that are consistent with what is observed in humans following TBI. The enlargement of the lateral ventricles that we observed histologically following TBI is also observed in human TBI patients, and it correlates with outcome; though delayed enlargement is associated with prolonged coma, the acute enlargement that we observed is typically indicative of cerebrospinal fluid obstruction due to hemorrhage and/or edema [96,97,98,99]. Periventricular white matter pathology is also common to closed-head moderate-to-severe TBI in humans and pigs, and the extensive axonal injury in the fornix/septum pellucidum that we observed is associated with cognitive deficits in humans following TBI, as well as neurodegenerative diseases such as Alzheimer’s, and can even predict cognitive decline in healthy adults [5,100,101,102,103,104]. The post-traumatic epilepsy and coma observed in our pigs are also frequently observed following moderate-to-severe TBI in humans, and importantly, the wakefulness without awareness that we observed in one of our TBI subjects is a human manifestation of TBI that, to our knowledge, has not previously been reported preclinically. While we are not drawing any conclusions based on these observations per se, they suggest that future studies in this model will allow us to test new clinically relevant hypotheses that were previously difficult or impossible to test preclinically.

While this study focused on acute critical care, we are also developing these capabilities to provide a path to recovery to enhance the translational relevance of rehabilitation and regenerative medicine research. Therefore, we monitor for signs of emergence from coma. During this study, we learned that merely emerging from a coma (i.e., responsive to noxious stimuli) may not indicate that the animal will soon be able to be extubated, become ambulatory, and resume self-feeding. Indeed, it appears that after high-rate sagittal angular rotational acceleration/deceleration of the head in pigs, animals may linger in a wakeful/unaware state for some time after emerging from a coma prior to regaining full awareness of their environment. This emphasizes the need for additional reverse translational development to adopt aspects of post-ICU care. We are currently consulting with physiatrists specializing in rehabilitative care for traumatic disorders of consciousness to establish protocols for administering post-ICU care to wakeful/unaware pigs after severe TBI. This extended post-coma, pre-recovery state will require replication, characterization, and unique care considerations, but the initial observation raises the exciting prospect of opening the study of traumatic disorders of consciousness to translational research [36].

This study was focused on technique development and optimization, and future studies will be needed to further characterize these injury manifestations, their similarities to those observed in humans, the mechanisms involved, and potential mitigation strategies. Furthermore, there are several modalities that were not employed in this study due to our focus on optimizing critical care. Most notably absent is neuroimaging due to our reluctance to transport subjects under these conditions during the critical care development phase. In other ongoing studies, we are utilizing sequences for in vivo T1, T2, susceptibility weighted imaging (SWI), and diffusion tensor imaging (DTI), as well as high-resolution ex vivo DTI sequences that have been validated in swine. Neuroimaging offers rich, vital data for brain trauma research and is a highly relevant clinical correlate, and, as such, it will be incorporated into upcoming studies utilizing this swine neurocritical care and monitoring platform. 

We recognize the current bottleneck that exists due to the highly specialized equipment and expertise necessary to work in this model, as well as the significant demands placed upon resources and personnel. Therefore, we will make every effort to share our data from future studies by tabulating compatible data for upload to platforms such as the Open Data Commons for TBI (ODC-TBI) [105]; scanning slides via Aperio for upload to eSlide manager; and biobanking FFPE brain blocks, as well as blood, CSF, and other samples, for future analyses with collaborators. We will also continue to advocate for funding agencies to work with research institutions and provide the necessary startup funds and resources to qualified researchers to expand the use of swine models, specifically the swine model of rotational-acceleration-induced TBI. Ultimately, engagement and investment from funding agencies and research institutions will be necessary to relieve this bottleneck in the translational neurotrauma pipeline.

### 4.1. Future Directions for Translational SAH and TBI Studies 

With the development of these techniques and the observations made along the way, there are many new avenues of investigation opened before us. We hope to utilize these capabilities for future studies of SAH, exploring the role of complement activation in vasospasm following injury, along with the pathological mechanisms linking SAH and stress cardiomyopathy. While this is not the only model of SAH, the inclusion of multimodal neuromonitoring and critical care enhances both the translational relevance and the depth of clinically relevant data collected. Future studies of TBI with coma will begin with investigating pathological correlates of coma severity, in particular, the pontine projections of the reticular activating system. The swine model of rotational-acceleration-induced TBI is a well-characterized model that has generated significant advancements in neurotrauma for decades. However, without the sophisticated neurocritical care afforded to human patients, moderate-to-severe TBI is very difficult to manage and often fatal. Therefore, with the humane treatment of research animals as a top priority, chronic studies have been limited to mild TBI [55,75,106], and studies of swine in extended coma have not exceeded 8 h [37,42,43,44]. Only by providing these subjects with the same neurocritical care afforded to a human patient can we ethically and practically extend beyond the acute period for translational study of rehabilitation [107]. Therefore, in addition to developing new diagnostics, therapeutics, and improving prognostic capabilities in the neurocritical care space, we can now do the same for the recovery/rehabilitation phase following moderate-to-severe TBI with coma.

### 4.2. Conclusions

Large animal models are both resource- and labor-intensive, and the inclusion of SAH or head rotational acceleration injury and neurocritical care significantly increases the specialized training and expertise required to have success. However, by establishing a multidisciplinary team, we demonstrated that it is feasible to replicate the mechanisms and manifestations of severe human neurotrauma in a large animal model with the inclusion of neurocritical care and monitoring. The preclinical study of SAH or TBI with neurocritical care requires an animal model that recapitulates injury mechanisms (e.g., head rotational loading) and manifestations (e.g., coma and increased ICP) observed in humans to bridge the translational gap between rodent studies and clinical trials. The integration of neuromonitoring and critical care into a model that uniquely recreates the forces of human TBI to create true moderate and severe TBI with coma further increases translational relevance and allows for preclinical study of the unique neurocritical care environment, while also extending the study period for moderate-to-severe TBI with coma and offering a path toward the preclinical study of traumatic disorders of consciousness.

## Figures and Tables

**Figure 1 biomedicines-11-01336-f001:**
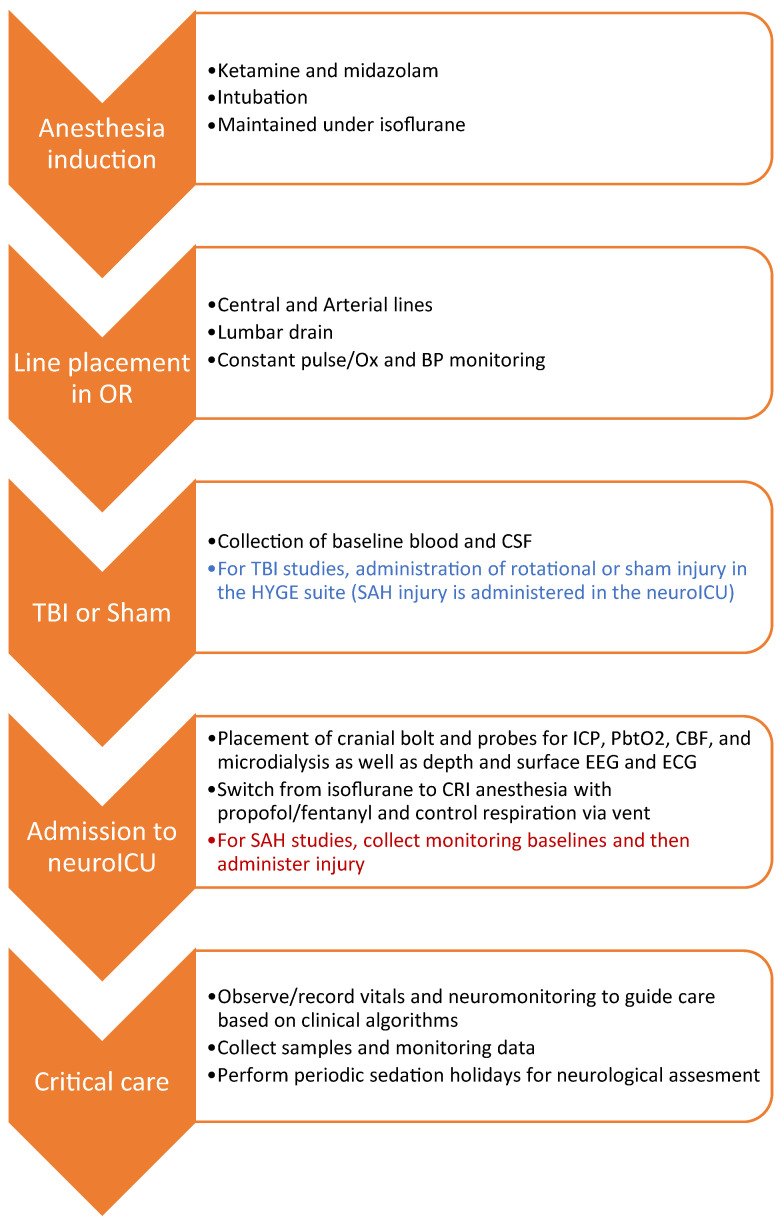
**Workflow for SAH and TBI in the swine neuroICU.** This flowchart details the order of events for SAH and TBI experiments utilizing neurocritical care and monitoring in the swine neuroICU.

**Figure 2 biomedicines-11-01336-f002:**
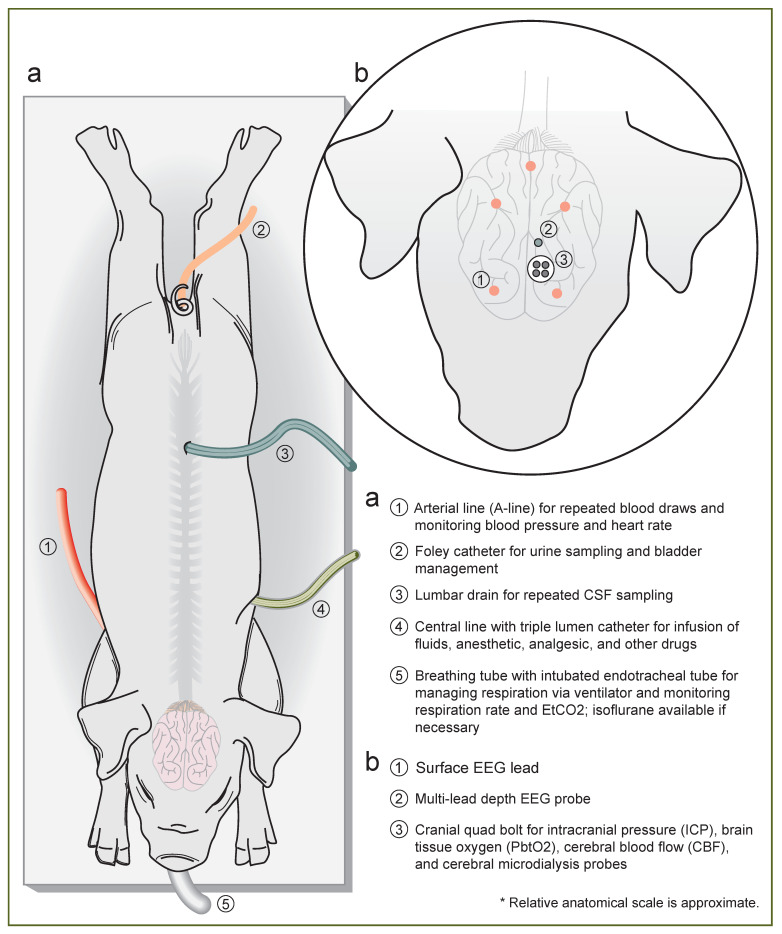
**Instrumentation and line placement in the swine neuroICU.** (**a**) Illustration of a pig in the neuroICU with lines labeled; (**b**) a close-up drawing of cranial monitoring in the swine neuroICU, with probes labeled. Artwork by Paul Schiffmacher.

**Figure 3 biomedicines-11-01336-f003:**
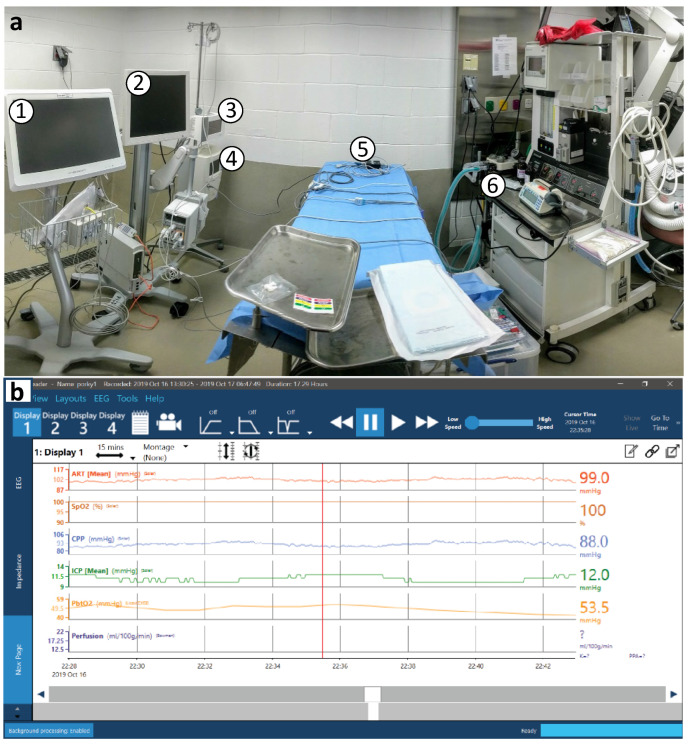
**Panoramic view of the swine neuroICU.** (**a**) Numbers indicate a few key pieces of equipment pictured in this panoramic view of the swine neuroICU. (1) Moberg CNS Monitor for time-synchronizing continuous EEG with systemic physiology from over 30 ICU monitoring devices. (2) GE Solar monitor for core vital signs. (3) Licox PbtO_2_ and temperature sensor. (4) Bowman Hemedex thermal diffusion cerebral perfusion monitor for CBF. (5) Various probes atop a variable pressure pad on the patient table. (6) Anesthesia and respiratory support. (**b**) A screen-capture image of example traces from the Moberg CNS Monitor during hour 22 of a swine neuroICU experiment (cerebral perfusion monitor cycles on and off by design).

**Figure 4 biomedicines-11-01336-f004:**
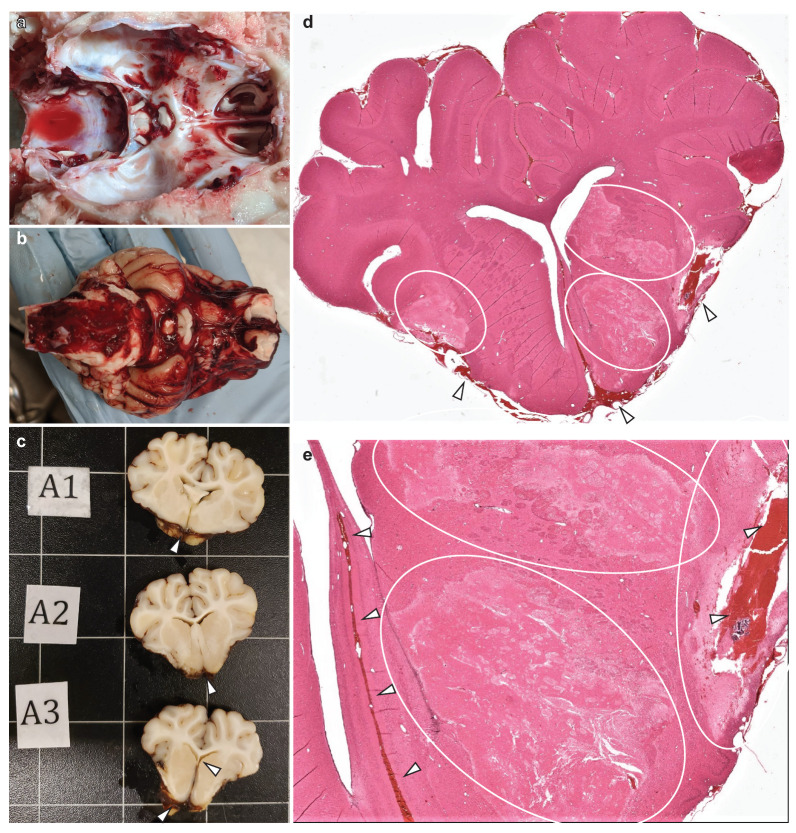
**SAH gross and histopathology.** Upon brain extraction, clear confirmation of blood deposition at the skull base was observed (**a**,**b**). Confirmation of blood deposition (white arrows) on the ventral brain surface and in the parenchyma was also confirmed during gross pathology (**c**). With H&E staining, blood (white arrows) can be found in sulci, fissure, and parenchyma, with pale staining indicative of ischemia (white circles) adjacent to the deposited blood (**d**,**e**).

**Figure 5 biomedicines-11-01336-f005:**
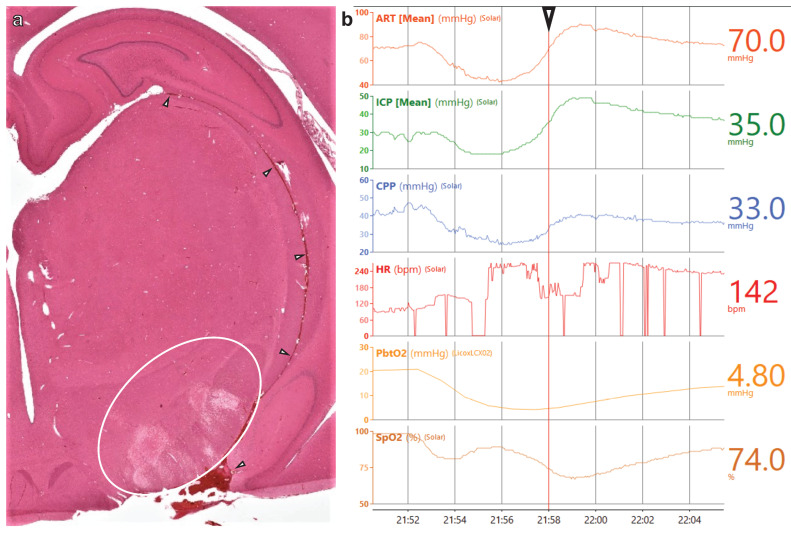
**SAH shock myocardia.** H&E staining reveals an ischemic area adjacent to blood at the base of the brain (white circle), with white arrows tracking blood traveling through a fissure to the hippocampus (**a**). Multimodal monitoring from the Moberg surrounding the time of the cardiac event (**b**). White arrow and vertical red line indicate the point on the traces for which specific values are provided on the right.

**Figure 6 biomedicines-11-01336-f006:**
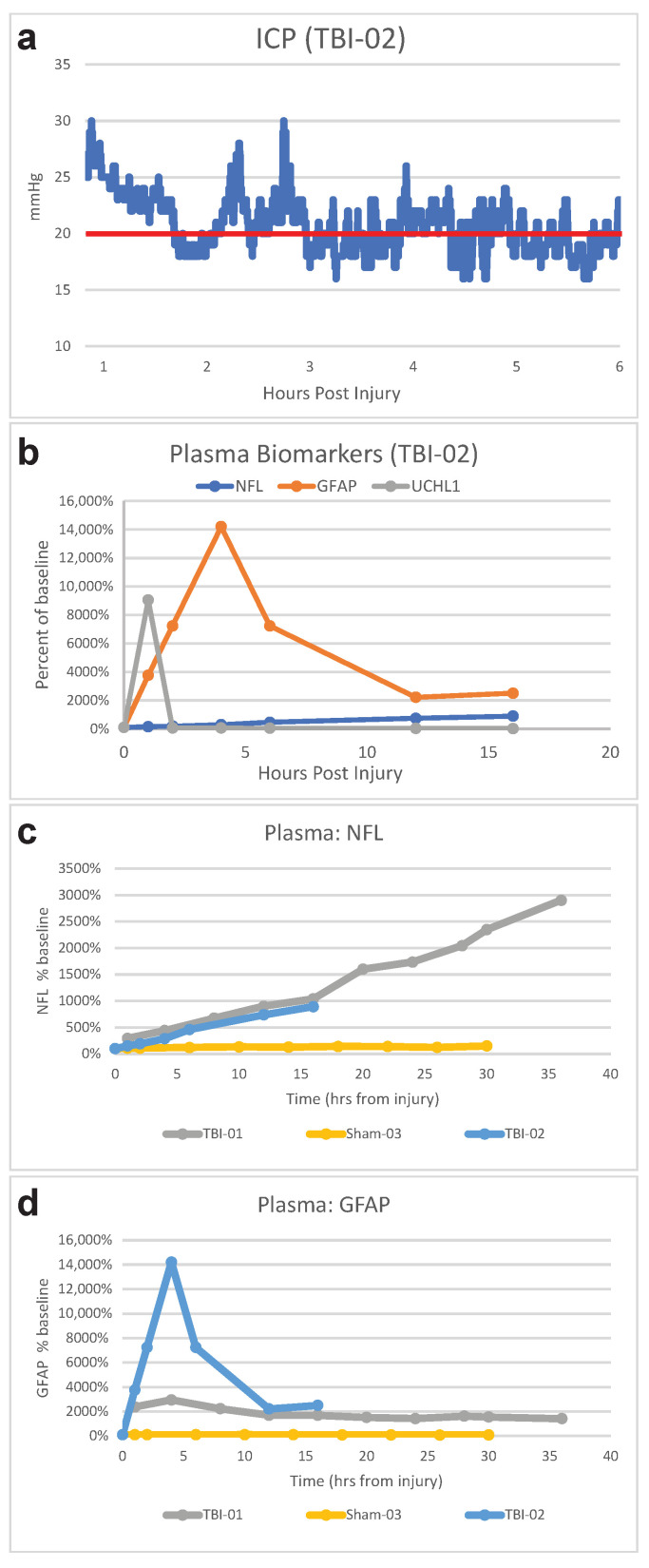
**TBI ICP and biomarkers.** Continuous ICP recording for 6 h following moderate-to-severe TBI with coma, with a red horizontal line to indicate the target of maintaining pressure below 20 mmHg (**a**). Results for the same animal demonstrating compatibility of swine plasma with a standard human TBI biomarker panel with NfL, GFAP, and UCHL1 (**b**). Plasma-biomarker-assay results for two TBI animals and a Sham animal, showing changes in NfL (**c**) and GFAP (**d**) after injury.

**Figure 7 biomedicines-11-01336-f007:**
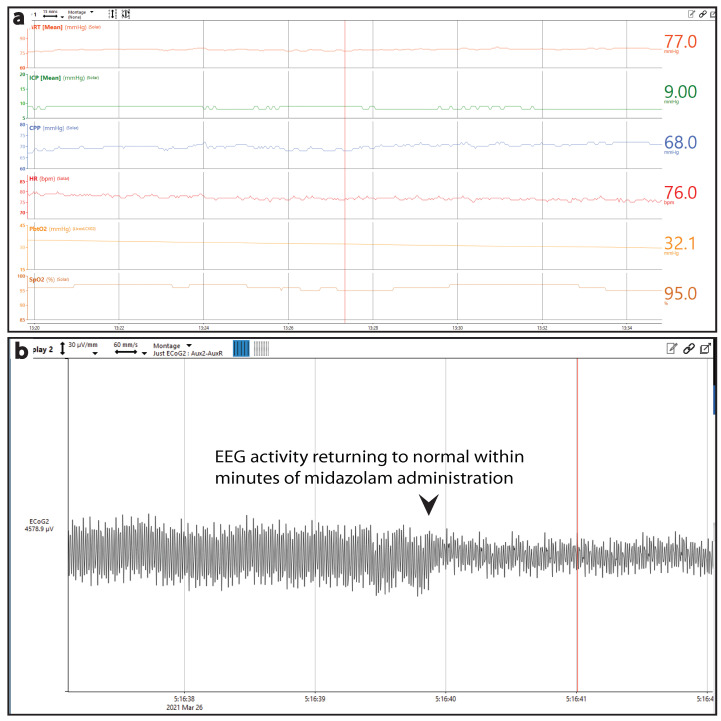
**Post-traumatic epilepsy following TBI with coma.** Multimodal monitoring during a 15-hour period of relative stability following early apnea and cardiac events (**a**). Single EEG trace showing termination of seizure immediately following midazolam administration (**b**).

**Figure 8 biomedicines-11-01336-f008:**
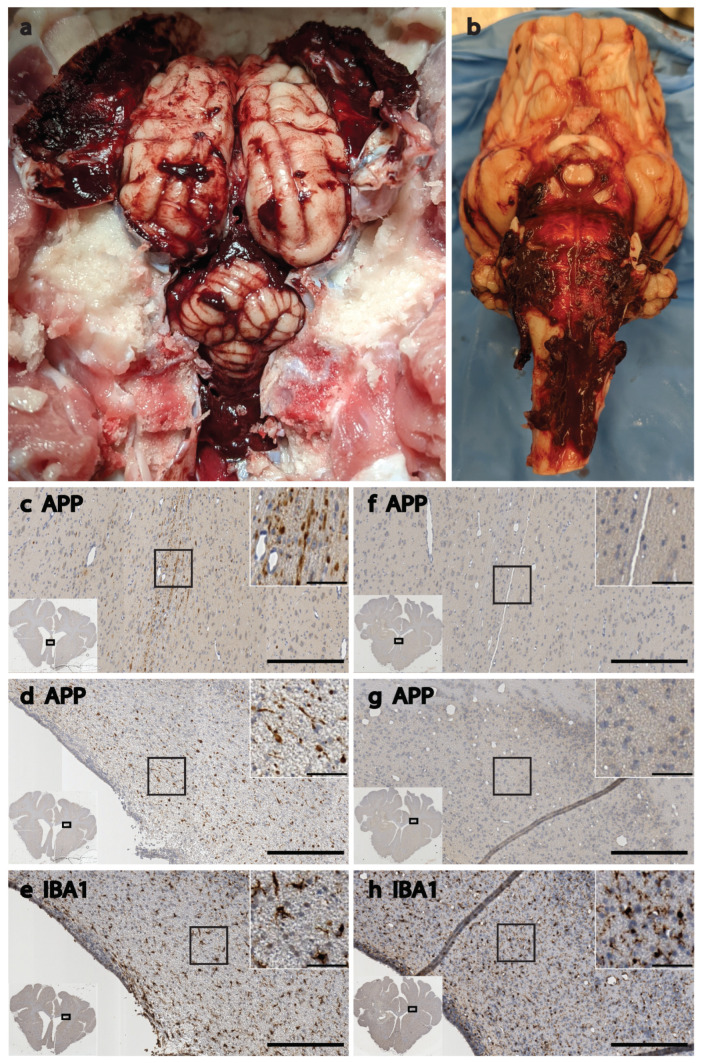
**General pathology following TBI with coma and prolonged emergence.** Extensive subdural bleeding was present during brain extraction, with clotting evident on the underside of the dura, on the brain’s surface, in the tentorium, and surrounding the brainstem (**a**,**b**). The same post-TBI brain is pictured in panels (**a**,**b**), and histology from this brain is pictured in panels (**c**–**e**). (**c**,**f**) Colorimetric histological images from the medial periventricular white matter comprising the fornix and septum pellucidum. (**d**,**e**,**g**,**h**) Images from the dorsolateral periventricular white matter. Each panel contains a large 15× image with an overlay in the bottom left showing the entire section, with a rectangle to indicate the region from which the 15× image was acquired, as well as a magnified view of cellular features in a call out box in the top right, also with a square on the 15× image to indicate the region being magnified. Enlarged ventricles are apparent in the TBI brain as compared to the sham brain, as shown in the full section insets. APP burden was scored as 3 out of 3 by a blinded analyst in the TBI brain (**c**,**d**), with no detectable APP pathology in similar sections from the sham brain (**f**,**g**); IBA1 stain reveals reactive microglia with short, thick processes in the TBI brain (**e**), while microglia in the sham brain (**h**) appear to have more ramified networks of processes extending out into the surrounding area, moving in and out of plane to create a speckled staining pattern in the parenchyma; large scale bars = 250 µm, and smaller inset scale bars = 50 µm.

**Table 1 biomedicines-11-01336-t001:** Examples of neuromonitoring in swine neurotrauma research.

Authors, Year	Title	Injury	Time	Anesthesia	Neuromonitoring Modalities	EEG	Other	A Line	Central Line	Lumbar Drain
Friess et al., 2011 [42]	“Neurocritical Care Monitoring Correlates with Neuropathology in a Swine Model of Pediatric Traumatic Brain Injury”	rotational TBI	6 h	isoflurane and CRI fentanyl	ICP, PbtO_2_, microdialysis for LPR	N	IHC	Y	Y	N
Friess et al., 2012 [43]	“Early cerebral perfusion pressure augmentation with phenylephrine after traumatic brain injury may be neuroprotective in a pediatric swine model”	rotational TBI	6 h	isoflurane and CRI fentanyl	ICP, PbtO_2_, CBF, microdialysis for LPR	N	IHC	Y	Y	N
Weenink et al., 2012 [47]	“Quantitative electroencephalography in a swine model of cerebral arterial gas embolism”	arterial gas embolism	4 h	IV ketamine, sufentanil, midazolam, and pancuronium bromide	ICP, PbtO_2_, microdialysis for lactate and glucose	Y (surface)	n/a	Y	Y	N
Nyberg et al., 2014 [45]	“Metabolic Pattern of the Acute Phase of Subarachnoid Hemorrhage in a Novel Porcine Model: Studies with Cerebral Microdialysis with High Temporal Resolution”	SAH	135 min	CRI ketamine, morphine, and rocuronium bromide	ICP, microdialysis for glucose and LPR	N	CT scan after experiment	Y	Y	N
Friess et al., 2015 [44]	“Differing effects when using phenylephrine and norepinephrine to augment cerebral blood flow after traumatic brain injury in the immature brain”	rotational TBI	6 h	CRI midazolam and fentanyl	ICP, PbtO_2_, CBF, microdialysis for LPR	N	IHC	Y	Y	N
Chen et al., 2017 [48]	“Quantitative electroencephalography in a swine model of blast induced brain injury”	blast TBI	2 h	IV propofol	none	Y (surface)	n/a	N	N	N
Mader et al., 2018 [49]	“Evaluation of a New Multiparameter Brain Probe for Simultaneous Measurement of Brain Tissue Oxygenation, Cerebral Blood Flow, Intracranial Pressure, and Brain Temperature in a Porcine Model”	CCI, physiological challenges	~5 h	CRI thiopental and piritramide	testing single probe for ICP, PbtO_2_, CBF	N	n/a	Y	Y	N
Datzman et al., 2019 [46]	“In-depth characterization of a long-term, resuscitated model of acute subdural hematoma–induced brain injury”	ASDH	54 h	CRI propofol and fentanyl	ICP, PbtO_2_, microdialysis for lactate and glucose	N	mGCS; IHC; brain tissue mitochondrial respiration (Oroboros); plasma GFAP and NSE	Y	Y	N
Cralley et al., 2022 [51]	“Zone 1 REBOA in a combat DCBI swine model does not worsen brain injury”	dismounted complex blast injury (DCBI)	6 h	CRI propofol and fentanyl	ICP	N	brain water content, MAP	Y	Y	N
Adedipe et al., 2022 [52]	“Left Ventricular Function in the Initial Period After Severe Traumatic Brain Injury in Swine”	fluid percussion injury	8 h	isoflurane	ICP	N	transesophageal echocardiography, coagulation, blood flow	Y	Y	N
Abdou et al., 2022 [53]	“Characterizing Brain Perfusion in a Swine Model of Raised Intracranial Pressure”	raised ICP via intracranial Fogarty balloon	2 h	isoflurane	ICP	N	computed tomography perfusion for CBF	Y	Y	N

**Table 2 biomedicines-11-01336-t002:** Swine Coma Scale scores during TBI-induced coma and prolonged emergence.

**To maximize arousal (5 min after stopping propofol):** grip and roll between fingers; first cheek, then neck (sternocleidomastoid), then shoulder/back (trapezious)						
**Swine Coma Scale - MAXIMIZE AROUSAL PRIOR TO EXAM -**		At each timepoint, indicate the score for each category.				
**Score**	**EYE BLINK**	**Baseline**	**2 h**	**4 h**	**8 h**	**12 h**
**3**	Blinks spontaneously (wait 3 min)	3			3	3
**2**	Blinks upon stimulation (e.g. pinch or ear tickle; not near eye)		2	2		
**1**	No blinking (without directly touching the eye)					
	**MOTOR ACTIVITY**					
**6**	Voluntary walking	6				
**5**	Sitting					
**4**	Isolated spontaneous movements (e.g. limbs or head)		4	4	4	4
**3**	Withdraws forepaw and/or hindpaw in response to noxious stimulation					
**2**	Muscle contractions in response to noxious stimulation of the limbs					
**1**	Absence of motor response to noxious stimulation					
	**AUDITORY RESPONSE**					
**2**	Auditory startle	2			2	2
**1**	None		1	1		
	**BRAIN STEM REFLEXES (score 3 and skip if motor score is 6)**					
**3**	Both palpebral AND pinna reflexes present	3	3	3	3	3
**2**	Palpebral OR pinna reflex present					
**1**	Absence of palpebral and pinna reflexes					
	**RESPIRATION**					
**4**	Not on ventilator, breathes with a regular pattern	4			4	4
**3**	Not on ventilator, breathes with an irregular pattern			3		
**2**	Breathes above ventilator rate (initiates spontaneous breaths)		2			
**1**	Breathes at ventilator rate or apnea (no spontaneous breaths)					
	**TOTAL**	18	12	13	16	16

## Data Availability

The data presented in this study are available upon reasonable request.

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
