# Peer review of "Multimodal Neuromonitoring and Neurocritical Care in Swine to Enhance Translational Relevance in Brain Trauma Research"

_biomedicines, 2023, doi:10.3390/biomedicines11051336_

Round 1
Reviewer 1 Report
Manuscript “Multimodal Neuromonitoring and Neurocritical Care in Swine to Enhance Translational Relevance in Brain Trauma Research” described some neurocritical care techniques and medical management of swine following subarachnoid hemorrhage and traumatic brain injury (TBI) with coma. The involvement of a multi-disciplinary team of neuroscientists, neurointensivists, and veterinarians ensured the clinical neuroICU and critical care pathways for the use in swine TBI model. Considering many similarities between swine and human in relation to the size and the structure of the brain this study is no doubt a key translational step forwards for the critical care of severe human TBI patients. The focus of the manuscript was to validate the establishment of a neurocritical care paradigm for acute moderate-to-severe swine TBI model and this was described in detail in the Methods part although extending to a chronic stage would be also important. In general, the experiments were carefully designed and performed, and the results were reliable based on the data presented in the manuscript. I only have some minor issues that need to be considered or addressed.
1. I couldn’t see and thus couldn’t download the list of abbreviations (Supplementary Table 1). I suggest including the list as a footnote in the manuscript. I also noticed that in some places the abbreviations were used before a whole name was given (e.g., CPP appeared first in page 5, but its complete name was found in page 17). Please check the similar situations for the entire manuscript.
2. Table 1 in page 5 should have a title (Table 1 ……). I suggest using the same citation style as in the main manuscript rather than write the titles for all the citations in the table, which occupies a large space. A title is also needed for Table 2.
3. The text in most of the figures (e.g., Figure 3, 5, 6, and 7) was small and the resolution was low. Please improve it.
4. There are some problems with Figure 8. First, it is not clear whether panels a and b were from a same brain (its dorsal and ventral side) or from two different brains. The sections from c to h were apparently from a TBI brain and a sham brain. Second, the resolution of the pictures was low. The insets in the upper right corner were not clearly demarcated. Third, without a quantitative data it is difficult to interpret the results presented by the figure even though it seems there were differences for different markers between the TBI and sham brain. Forth, the purpose to use APP and IBA1 for immunostaining was not stated. It was even not explained as what these two markers stain in the Methods.
Author Response
We are sincerely grateful for your time and your recognition of this model’s strengths and potential. We agree that more chronic studies are needed, and we are currently engaged in that pursuit. We apologize for the resolution and formatting issues you encountered as described in comments 1-3. Unfortunately, the tables and figures included within the word document for review are not representative of the quality that will be present in the final publication. However, we can assure you that these formatting/resolution issues will be resolved (no pun intended) with upload of the full-sized files for proofing. We have added the supplementary abbreviation table to the end of the word document and corrected all term/abbreviation order of appearance issues. We will incorporate your title and citation comments for Tables 1 and 2 into the final file submitted for proofing. Similarly, we are making your suggested changes to Figure 8, which will appear in the final, full-sized version of the file submitted for proofing. Finally, we were surprised to realize that we had not described the reasons for staining APP and IBA1, and grateful for your catch. We have included explanations for the use of these markers in assessing brain pathology.
Reviewer 2 Report
The advantage of the paper is reporting the ins and outs of the development of a system enabling to provide a prolong neuromonitoring and intensive care in a swine models the popularity of which has been increased last years. Some drawbacks are exist: (1)The inclusion of technical details, however, may be arranged as a supplement (some figures or tables). (2) The conclusion of the paper looks overly broad; no direct advantages/disadvantages of the proposed system versus data that have describe neurocritical care studies in a similar (swine) model (brain trauma, stroke modeling etc.) are shown. (3) Too small number of most recent papers (2021-2022) directly related to a neurocare in a swine model are discussed (for example, Cralley, Moore, Fox et al., 2022Adelipe et al., 2022; Abdou et al., 2022).
Author Response
As stated with Reviewer 1, we are sincerely grateful for your time and your recognition of this model’s strengths and potential. We agree that typically the degree of technical detail in our methods would be reserved for a supplemental section. However, in this case we hope that the manuscript can go beyond reporting our own experiences and help others set up similar models, so we would like to maintain the attention to detail in the methods section. We have reigned in our concluding statements to clarify that the greatest advantage of this workflow is the ability to study neurocritical care in the context of true moderate and severe TBI with coma, which is currently unique to our lab. Finally, we are grateful for the more recent swine neuromonitoring references that you provided, and have added the citations to the paper and the final version of the table that will be included in proofing (though the final table is not included in the review word document).